# Design of Fuzzy PID Controller Based on Sparse Fuzzy Rule Base for CNC Machine Tools

**Zaiqi Yu** [1] , **Ning Liu** [1,*] , **Kexin Wang** [2] , **Xianghan Sun** [1] **and Xianjun Sheng** [1]

1 School of Electrical Engineering, Dalian University of Technology, Dalian 116024, China
2 School of Mechanical Engineering, Dalian University of Technology, Dalian 116024, China
* Correspondence: liun@dlut.edu.cn; Tel.: +86-152-4260-2608

**Abstract:** The robustness of the control algorithm plays a crucial role in the precision manufacturing and measurement of the CNC machine tool. This paper proposes a fuzzy PID controller based on a sparse fuzzy rule base (S-FPID), which can effectively control the position of a nonlinear CNC machine tool servo system consisting of a rotating motor and ball screw. In order to deal with the influences of both the internal and external uncertainties in the servo system, fuzzy logic is used to adjust the proportion, and integral and differential parameters in real-time to improve the robustness of the system. In the fuzzy inference engine of FPID, a sparse fuzzy rule base is used instead of a full-order fuzzy rule base, which significantly improves the computational efficiency of FPID and saves a lot of RAM storage space. The sensitivity analysis of S-FPID verifies the self-tuning ability of its parameters. Furthermore, the proposed S-FPID has been compared with the PID and FPID via simulation and experiment. The results show that compared with the classical PID controller, the overshoot of the S-FPID controller is reduced by 74.29%, and the anti-interference ability is increased by 62.43%; compared with FPID algorithm, the efficiency of the SPID is improved by 87.25% on the premise of a slight loss in robustness.

**Keywords:** CNC machine tool; fuzzy PID control; nonlinear control; adaptive control

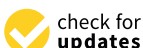



## 1. Introduction

With the development of a computer numerical control (CNC) system, precision machining and measurement have become a pervasive technology today [1]. In order to ensure the quality of products, the research of its control algorithm has important theoretical significance and practical value. The factors that affect the precision of CNC equipment include tracking error and multi-axis coupling error [2]. Due to the uncertainty of internal parameters and the disturbances of the external environment, the tracking control in the servo system is a nonlinear problem [3] that will decrease the robustness of the system.

The classical PID controller is still the most widely used control method in motion control systems due to its mature theory, high reliability, high control accuracy, and simple structure [4]. However, the PID controller designed using traditional typical methods needs an accurate linear mode, which is difficult to apply in a nonlinear systems [5]. A big disadvantage of PID consists of the computing of the controller gains [6]. In order to achieve a quick dynamic response, the nonlinear dynamic model must be considered in the design of the controller [7]. Modern control and other techniques such as adaptive and artificial intelligence control for changing gains are currently used as alternatives for adaptation mechanisms to system changes in time [8]. Numerous approaches are proposed in the literature to solve the dynamic response and robust problem of nonlinear systems based on various control schemes, including nonlinear disturbance observer-based control algorithm (NDOBC), nonlinear PID, sliding mode controller, fuzzy control algorithm, and so on. The NDOBC presented in [9] estimates the fast time-varying disturbances of the spacecraft system, and attenuates its influence on the control system through feedforward

compensation, which improves the robust dynamic performance and attitude tracking accuracy. Furthermore, considering the high degree of iteration, complex structure, and high dependence on the accuracy of the control system modeling, the observer-based disturbance prediction algorithm is difficult to apply. The nonlinear PID controller is the earliest discovered and most commonly used method to solve nonlinear problems. However, due to the use of the nonlinear PID controller, most of them have increased numbers of turning parameters, and this will impose a burden on industrial operators for their tuning [10]. Although some scholars have proposed better adaptive parameter adjustment methods, such as the adaptive adjustment of nonlinear parameters through fuzzy logic [11] and the application of saturation function to nonlinear PID control law [12], they also bring extremely high complexity to the controller algorithm. The sliding mode controller makes the system have an invariance that is superior to the robustness when it moves on the sliding mode surface. However, it has a chattering phenomenon in the steady state because of the variable structure [13]. The fuzzy logic system is an inference system to mimic human thinking, and the experience of human engineers is implemented in the control algorithm in the form of IF-THEN rule statements through a fuzzy inference engine [14]. Common fuzzy controller structures are as follows: traditional fuzzy controller, adaptive fuzzy controller, fuzzy control combined with other control algorithms [15], etc. Fuzzy control combined with other control algorithms includes fuzzy PID (FPID), fuzzy sliding mode control (FSMC), neuro-fuzzy Control (NFLC), and so on, and especially, FPID is usually used. For example, the fuzzy self-turning PID controller proposed in [16] was used to control the temperature in the waste heat recovery system, based on the Rankine cycle. A novel adaptive FPID controller is designed in [17] for geostationary satellite attitude control. The model predictive and fuzzy logic control theory proposed in [18] is used to study the optimal control of the pumped storage unit (PSU) under a no-load start-up condition at a low head area. However, the above FPID controllers based on the full order fuzzy rule base not only need a large amount of RAM storage space in the fuzzy logic operation, but they also have multiple cycle operations with a fast frequency in the motion control application, which requires a harsh performance of the CPU.

This paper proposes a kind of FPID based on sparse fuzzy rule base (S-FPID), which has the following advantages: (1) Compared with the classical PID controller, S-FPID has a higher robustness for the servo system, with both internal and external disturbances. (2) Due to the use of a sparse fuzzy rule base instead of a full-order fuzzy rule base in the fuzzy inference engine, it can obviously improve the operation efficiency at the cost of a less robust performance loss. (3) It can make the application of the micro CPU with weak performance possible, which will not only save the cost, but also reduce the quality of the equipment. (4) It can also improve the smoothness of the operating system.

The rest of the paper is organized as follows. The dynamic model of the servo system is presented, and the uncertainty and control problems of the system are analyzed in the next section. In Section 3, the S-FPID controller is designed, and its self-tuning sensitivity of the control parameters is analyzed. In Section 4, the performance of S-FPID is verified via simulation and experiment. A summary of the work is given in Section 5.

## 2. Servo System Dynamics Model

In order to analyze the robust tracking control problem of the servo system affected by both the internal and external disturbance uncertainties, the dynamic modeling of the servo system is carried out in this section.

### 2.1. Servo System Reference Model

The servo system is mainly composed of a servo motor and ball screw. The encoder feeds the speed and angle information of the servo motor to the servo driver in real-time, and the servo driver controls the output torque of the motor by controlling the supply of power. The ball screw converts the rotational motion of the motor into the linear motion of the platform. Because of the high manufacturing precision of the ball screw, the servo motor

encoder can be used as the position feedback sensor, and the semi-closed-loop control method is adopted. Supplemented with an effective control algorithm, the control precision of the servo system can be satisfied.

As shown in Figure 1, in order to describe the dynamic model of the servo system explicitly, the mechanical structure and physical parameter symbols of the servo system are presented.

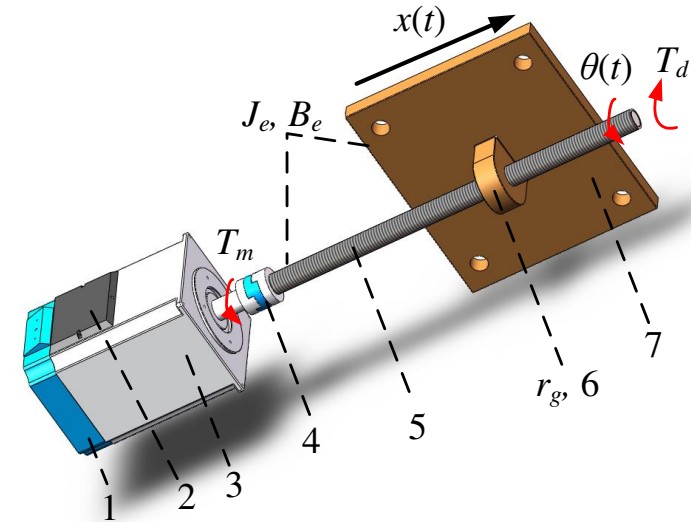

| 1 - Encoder | 2 - Power supply | 3 - Motor |
|---|---|---|
| 4 - Coupling | 5 - Screw | 6 - Nut |
| 7 - Platform | | |

**Figure 1.** Servo system reference platform.

$T_m$ is the output torque of the servo; $T_d$ is the equivalent disturbance torque of the servo system, which is mainly affected by the friction between the platform and guide; $x(t)$ and $\theta(t)$ are expressed as the displacement of the platform and the angular displacement of the ball screw, respectively; $J_e$ is the equivalent moment of the inertia of the servo system; $B_e$ is the equivalent rotational damping of the system; and $r_g$ is the transmission ratio of the ball screw.

### 2.2. Servo System Modeling

The output torque $T_m$ of the three-phase motor is approximately proportional to its supply voltage $u$ in the low-frequency domain, which can be expressed as:

$$T_m = K_a K_t u \tag{1}$$

where $K_a$ is the gain constant of the current amplifier and $K_t$ is the torque constant. $K_m = K_a K_t$ can be approximated as the gain of the servo motor output torque $T_m$ with respect to the input voltage $u$.

Due to the high mechanical precision of the servo system, the displacement $x(t)$ of the platform and the equivalent displacement $\theta(t)$ of the angular displacement of the ball screw can be expressed as follows:

$$x(t) = r_g \theta(t) \tag{2}$$

where the transmission ratio $r_g$ of the ball screw pair represents the lead $h$ corresponding to one rotation of the screw $(2\pi)$, that is, $\frac{h}{2\pi}$. According to the conservation of torque, the dynamic equation of the screw drive mechanism can be obtained as follows:

$$J_e \ddot{\theta}(t) = T_m - T_d - B_e \dot{\theta}(t) \tag{3}$$

By synthesizing Equations (1) and (3), the differential equation between the input $u(t)$ and output $x(t)$ of the servo system can be obtained. The transfer function expression of the servo system can be obtained via Laplace transform:

$$X(s) = \frac{r_g}{(J_e s + B_e)s}(K_m U(s) - T_d) \tag{4}$$

where the transfer function of the servo system is $\frac{r_g K_m}{J_e s^2 + B_e s}$, which is a typical seconder-order system.

### 2.3. Uncertainty and Control Problem Analysis

The uncertainty of the servo system comes from the internal parameters and the environmental disturbances. Taking the equivalent moment of inertia $J_e$ and the equivalent disturbance torque $T_d$ of the servo system as an example, their parameter values are no longer of fixed value, but they fluctuate in an interval, such as:

$$\begin{cases} J_e^* = J_{e0} \pm \triangle J_e \\ T_d^* = T_{e0} \pm \triangle T_e \end{cases} \tag{5}$$

where $J_e^*$ and $T_d^*$ denote the actual values of the moment of inertia and the disturbance torque, respectively; $J_{e0}$ and $T_{d0}$ denote the initial estimates of them, respectively. $\Delta J_e$ and $\Delta T_d$ denote the fluctuation interval of them, respectively. Substituting Equation (5) into Equation (4), we can obtain:

$$\left[ (J_{e0} \pm \Delta J_e)s^2 + B_e s \right] X(s) = r_g [K_m U(s) - (T_{d0} \pm \Delta T_d)] \tag{6}$$

It can be seen that the uncertainty of the moment of inertia acts on the quadratic term of the platform displacement, and the uncertainty of the interference torque directly acts on the input voltage. Under the accumulation of time, the control performance of the servo system will be adversely affected.

The control problem of the servo system can be summarized as follows: A predetermined platform trajectory $x_r(t)$ is given, and a robust tracking controller is designed for the servo system with a nonlinear problem, which can ensure that the position error of the platform $x_e = x_r(t) - x(t)$ can converge to zero quickly and stably asymptotically with time.

## 3. Servo Control Algorithm Design

At present, most motion control systems still use a PID controller. Furthermore, the PID controller cannot adjust the nonlinear system adaptively in real-time, which will cause a reduction in the dynamic response ability, and the stability of the system. Fuzzy logic control can be applied to the controller with the accumulated experience of human beings, which is an effective mathematical method for dealing with nonlinear problems [3]. FPID is a controller that combines fuzzy logic with the classical PID method, which not only has the robustness of the classical PID controller, but it also can solve the control problem of the servo system with uncertain factors in this paper [14].

### 3.1. FPID Controller for the Servo System

Figure 2 shows the overall structure of the FPID controller. The servo motor encoder feeds the displacement information of the platform, which forms a semi-closed loop struc-

ture, and it can track the given value in real-time. The current position error $x_e$ and its increment $x_{ec}$ are used as the input of fuzzy logic, and the PID controller parameter gains $\Delta K_p$, $\Delta K_i$, and $\Delta K_d$ are used as the output to adjust the PID controller parameters adaptively online. Finally, the motor voltage $u$ is outputted from the PID controller.

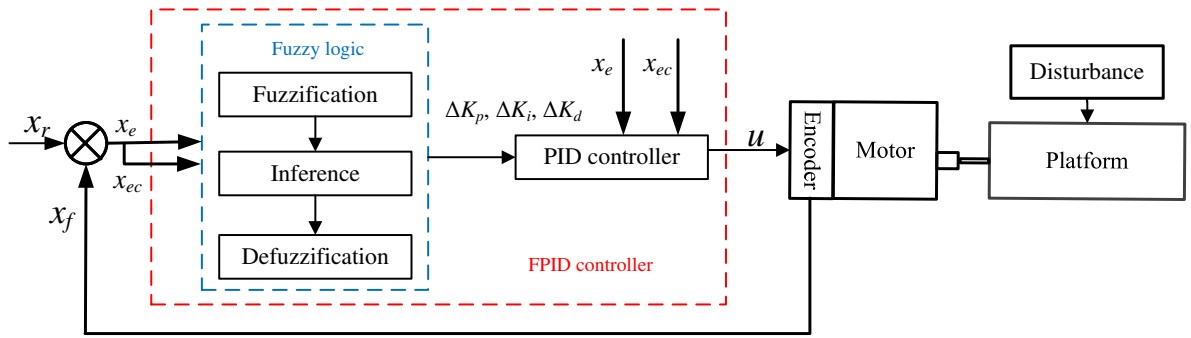

**Figure 2.** The structure of FPID.

Firstly, it is known that the discrete expression of the traditional PID controller is:

$$\begin{cases} u(k) = K_p e(k) + K_i \sum_{n=0}^{k} e(n) + K_d(e(k) - e(k-1)) \\ K_i = \frac{K_p \times T}{T_i}, K_d = \frac{K_p \times T_d}{T} \end{cases} \tag{7}$$

where $T$ represents the sampling time, $T_i$ represents the integration time, $T_d$ represents the differential time, and $e(k)$ represents the deviation of the $k_{th}$ sampling period. For the set value $x_r$ and feedback value $x_f$, the error and error increments are expressed as follows:

$$\begin{cases} x_e(k) = x_r(k) - x_f(k) \\ x_{ec}(k) = \frac{x_e(k) - x_e(k-1)}{T} \end{cases} \tag{8}$$

In order to overcome the influences of both internal and external disturbance uncertain parameters in the servo system adaptively, the fuzzy logic output parameter gains $\Delta K_p$, $\Delta K_i$, and $\Delta K_d$ in real-time to adjust the proportional, integral, and differential parameters in Equation (7), which has the same update frequency as $x_e$ and $x_{ec}$ in Equation (8), such as:

$$\begin{cases} K_p(k) = \Delta K_p(k) + K_{p0} \\ K_i(k) = \Delta K_i(k) + K_{i0} \\ K_d(k) = \Delta K_d(k) + K_{d0} \end{cases} \tag{9}$$

where $K_{p0}$, $K_{i0}$, and $K_{d0}$ indicate the initial PID parameters; $\Delta K_p$, $\Delta K_i$, and $\Delta K_d$ represent the PID parameter gains of fuzzy logic output in the current sampling period; $K_p(k)$, $K_i(k)$, and $K_d(k)$ indicate the value of the PID parameter after adjusting adaptively. It can be seen in Equation (7) that $K_p$, $K_i$, and $K_d$ affect the voltage output value $u$ from three aspects: error response, steady-state error, and error prediction. The adaptive PID parameter values participate in the discrete PID operation, which brings stronger robustness to the controller.

### 3.2. Implementation of the S-FPID Controller
#### 3.2.1. FPID Structure

Fuzzy logic can use mathematical methods to realize people's reasoning thinking. As shown in Figure 2, it consists of three parts: fuzzifier, a fuzzy inference engine, and a defuzzifier [14]. Fuzzifier converts the input variables into fuzzy sets to be reasoned with via membership function; In fuzzy inference engines, IF–THEN rules follow max–min composition to transform input fuzzy sets into output fuzzy sets, which are saved in the

form of a fuzzy rule base; The defuzzifier converts the inference fuzzy set into real numbers through the centroid method, and outputs them.

For IF–THEN rules Based On max–min Composition [19]: Let $A$, $A'$, and $B$ be fuzzy sets of $X$, $X$, and $Y$, respectively. Assume that the fuzzy implication $A \rightarrow B$ is expressed as a fuzzy relation $R$ on $X \times Y$. Then, the fuzzy set $B'$ induced by "$x$ is $A'$", and the fuzzy rule "if $x$ is $A$, then $y$ is $B$" is defined by:

$$\mu_{B'}(y) = \max_x \min[\mu_{A'}(x), \mu_R(x,y)]$$
$$= \vee_x [\mu_{A'}(x) \wedge \mu_R(x,y)] \tag{10}$$

or, equivalently:

$$B' = A' \circ R = A' \circ (A \rightarrow B) \tag{11}$$

Defuzzification using the centroid strategy [16]:

$$Y = \frac{\int_V \mu_{\delta'}(y) \cdot ydy}{\int_V \mu_{\delta'}(y)dy} \tag{12}$$

where $Y$ is the defuzzified real value, $\mu$ is the membership degree of fuzzy grade, $y$ is the fuzzy level, and $V$ is all of the fuzzy grade intervals.

### 3.2.2. Fuzzification

In order to simplify the operation and to shorten the control cycle, the membership function of the input variables is in a relatively simple form. As shown in Figure 3, the input variables error $x_e$ and error increment $x_{ec}$ are fuzzified into seven types of fuzzy sets through the triangle membership function. They are NB, NM, NS, ZE, PS, PM, and PB, respectively. These seven sets represent the magnitude of input variables, and the specific meanings are shown in Table 1. The independent variable interval of membership function is selected through experience according to the characteristics of the experimental object. Taking error $e$ as an example, when the error value is greater than the upper limit of 1000, the $K_p$ gain will not change any more, but it will maintain the maximum value to track the position at the fastest speed.

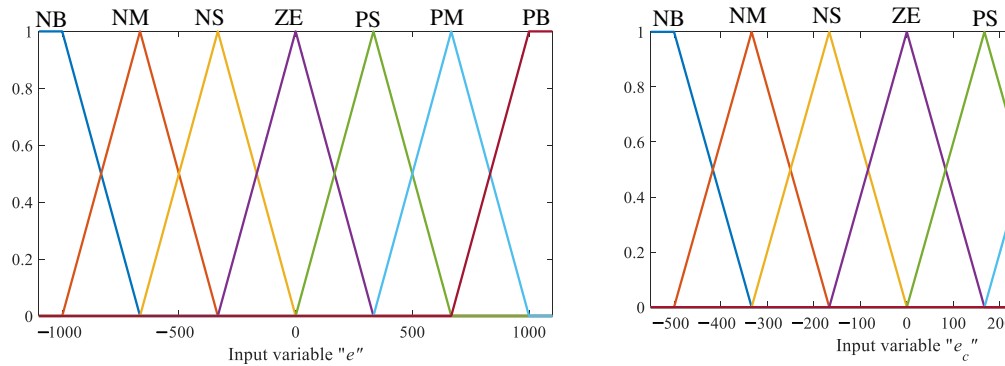

**Figure 3.** Membership function of input variables $e$ and $e_c$.

**Table 1.** Meanings of fuzzy subsets.

| Abbreviation | Meaning |
|---|---|
| NB | Negative large |
| NM | Negative medium |
| NS | Negative small |
| ZE | Zero |
| PS | Positive small |
| PM | Positive medium |
| PB | Positive large |

### 3.2.3. Fuzzy Inference Using a Sparse Fuzzy Rule Base

The adaptive fuzzy rule bases of the PID parameter gains $\Delta K_p$, $\Delta K_i$, and $\Delta K_d$ are shown in Tables 2–4, respectively. The first column of each table are the seven kinds of fuzzy subsets of error $x_e$, the first row represents the error gain $x_{ec}$, and the other cells represent the PID parameter gains derived from IF-THEN reasoning. However, some parameters in the fuzzy rule base have little effect on the servo system. For example, when the value of error $x_e$ is very large and the value of error gain $x_{ec}$ is also very large, the system is in a state of serious offset and accelerating away from the set point, which is rarely seen in the actual application of the servo system. Therefore, in order to reduce the CPU operation pressure and to release the RAM storage space, this paper designs the sparse fuzzy rule base instead of the original full-order fuzzy rule base, and the part with the yellow background in the table are the non-zero cells of sparse rule bases. In addition, some cells of the sparse rule bases are adjusted compared with those of the full-order bases, and the values in brackets are the changed values. This kind of FPID using a sparse fuzzy rule base in fuzzy inference engine is called S-FPID in this paper.

**Table 2.** Fuzzy rule base of $\Delta K_p$ (yellow background are the cells of sparse rule bases).

| $\Delta K_p$ | NB | NM | NS | ZE | PS | PM | PB |
|---|---|---|---|---|---|---|---|
| ine NB | 3 | 3 | 3 | 3 | 3 | 3 | 3 |
| NM | 3 | 3 | 3 | 3 | 2 | 2 | 2 |
| NS | 3 | 2 | 1 | 0 | −1 | −2 | −2 |
| ZE | 3 | 3 | 3 | 3 | 3 | 3 | 3 |
| PS | −2 | −2 | −1 | 0 | 1 | 2 | 3 |
| PM | 2 | 2 | 2 | 3 | 3 | 3 | 3 |
| PB | 3 | 3 | 3 | 3 | 3 | 3 | 3 |

**Table 3.** Fuzzy rule base of $\Delta K_i$ (yellow background are the cells of sparse rule bases).

| $\Delta K_i$ | NB | NM | NS | ZE | PS | PM | PB |
|---|---|---|---|---|---|---|---|
| ine NB | −3 | −3 | −3 | −3 | −3 | −3 | −3 |
| NM | −3 | −3 | −3 | −3 | −3 | −3 | −3 |
| NS | −3 | −2 | −1 | 0 | 1 | 2 | 3 (0) |
| ZE | 3 | 3 | 3 | 3 | 3 | 3 | 3 |
| PS | 3(0) | 2 | 1 | 0 | −1 | −2 | −3 |
| PM | −3 | −3 | −3 | −3 | −3 | −3 | −3 |
| PB | −3 | −3 | −3 | −3 | −3 | −3 | −3 |

**Table 4.** Fuzzy rule base of $\Delta K_d$ (yellow background are the cells of sparse rule bases).

| $\Delta K_d$ | NB | NM | NS | ZE | PS | PM | PB |
|---|---|---|---|---|---|---|---|
| ine NB | 3 | 3 | 3 | 3 | 3 | 3 | 3 (−3) |
| NM | 3 | 3 | 3 | 2 | 1 | 1 | 0 (−3) |
| NS | 2 | 2 | 1 | 0 | −1 | −2 | −2 (−3) |
| ZE | −3 | 1 | 2 | 3 | 2 | 1 | −3 |
| PS | −2 (−3) | −2 | −1 | 0 | 1 | 2 | 2 |
| PM | 0 (−3) | 1 | 1 | 2 | 3 | 3 | 3 |
| PB | 3 (−3) | 3 | 3 | 3 | 3 | 3 | 3 |

The fuzzy rule base is obtained by summarizing the human experience. For example: when the position error $x_e$ of the control system is very large and the error gain $x_{ec}$ is small, it proves that the system is in a state of deviation from the set value, but adjusting actively. It is necessary to increase $K_p$ and to decrease $K_i$ to increase the response speed of the system and to prevent the integral saturation of the system. When the position error $x_e$ of the control system is small and the error gain $x_{ec}$ is also small, the system is in a steady state. It is necessary to increase $K_p$ and $K_i$ to enhance the static stability of the system, increase the anti-interference ability of the system, and reduce the steady-state error of the system. The formulation of inference rules in the table of the servo system control can be interpreted as follows:

1. When the error is large: increase $K_p$ and $K_d$ to ensure the response speed; $K_i$ is set to zero to prevent integration saturation.
2. When the error is medium: reduce $K_p$ to prevent overshoot due to mechanical inertia.
3. When the error is small: increase $K_p$ and $K_i$ to enhance the stability and anti-interference ability of the system.
4. When the error gain is small: increase $K_d$, speed up the reaction speed of small error and resistance to small disturbance.
5. When the error gain is large: reduce $K_d$ to prevent the occurrence of jitter phenomenon.

### 3.2.4. Defuzzification

The real output value of the fuzzy controller is calculated using the centroid method. Taking the gains of $K_p$ as an example, Figure 4 shows its fuzzy control surface. The gain of $K_p$ is a nonlinear function of $x_e$ and $x_{ec}$, which indicates that the parameters of the PID controller can adjust adaptively. So, human knowledge and experience are implemented in the fuzzy rules successfully, which offers the S-FPID controller robust performance for the servo system with the nonlinear problem.

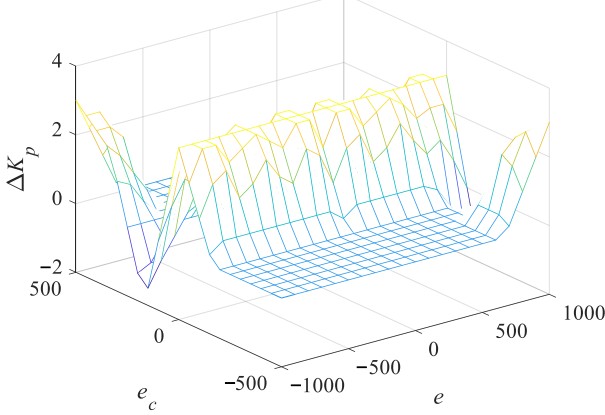

**Figure 4.** Fuzzy control surface of $\Delta K_p$.

### 3.3. Sensitivity Analysis

In order to verify the adaptive ability of the parameters of the S-FPID controller, a step response simulation experiment is designed for the servo system using MATLAB. Taking the PID parameter gains $\Delta K_p$ and $\Delta K_i$ as examples, the self-tuning process with respect to $x_e$ and $x_{ec}$ is observed in real-time, and the sensitivity is analyzed. Figures 5 and 6 show the curve of the proportion and integral parameters of the S-FPID controller with respect $x_e$ and $x_{ec}$, respectively. The tracking error value $x_e$ decreases from a given step of 1000 to 0, with an approximately constant slope, and $x_{ec}$ is the derivative of $x_e$. As shown in Figures 5 and 6, $K_p$ increases when the system deviates from the set value to ensure response speed, decreases when it approaches to the set value to prevent overshoot, and increases again when it reaches the steady state to ensure static stability. $K_i$ decreases when the system seriously deviates from the set value to prevent integral saturation, and increases gradually when approaching the set value, until it reaches the maximum in the steady state to ensure static stability. The S-FPID parameters are adaptively adjusted with inputs $x_{ec}$ and $x_e$, according to the rules in the experience base, and the self-tuning sensitivity is good, which can ensure that the S-FPID controller is still robust to the system affected by both the internal parameters and the external disturbance factors.

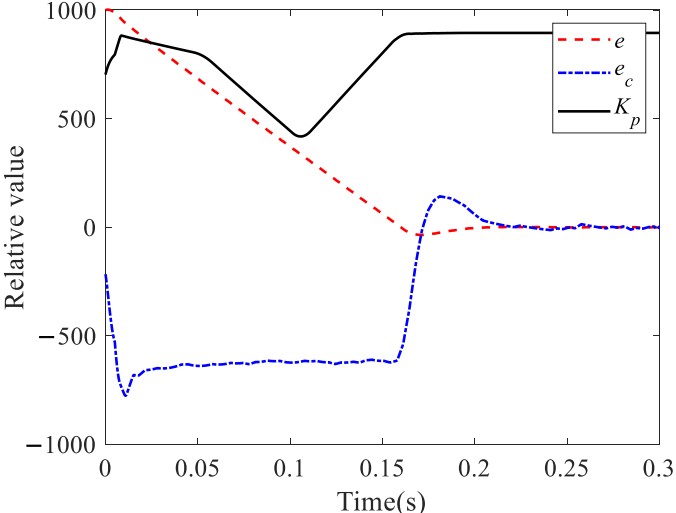

**Figure 5.** Adaptive sensitivity of $K_p$.

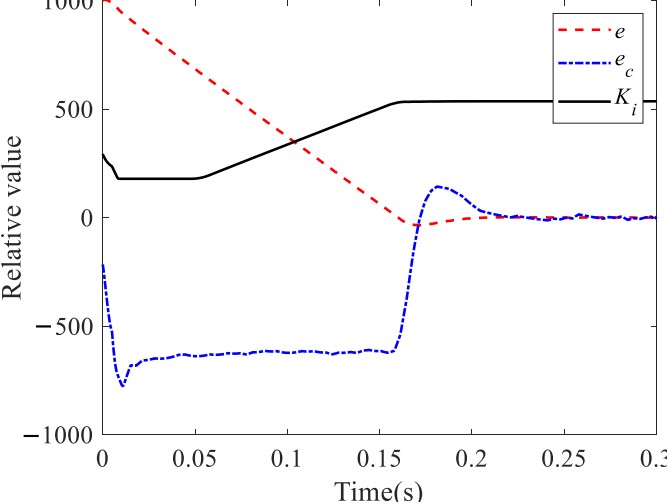

**Figure 6.** Adaptive sensitivity of $K_i$.

## 4. Experiment and Results

In order to verify the effectiveness of the S-FPID controller proposed in this paper, the position tracking control experiment is carried out via MATLAB simulation and servo physical experiment, and the tracking performances and anti-interference abilities of PID in the control card, FPID, and S-FPID control algorithms are compared and analyzed. In addition, the operation times of SPID and S-FPID fuzzy logic are compared in MATLAB.

### 4.1. Simulation and Analysis

As shown in Figure 7, the FPID controller module is built using the Simulink toolbox in MATLAB to control the speed of the servo motor. Figure 8 shows the PID controller model for Simulink. The value of the PID controller is output after passing through a limiting module, which can prevent the motor from burning due to excessive current. The integral limit is also set to prevent integral saturation. In practical application, by double clicking, the parameter values, output limits and sampling time of the PID module can be manually set.

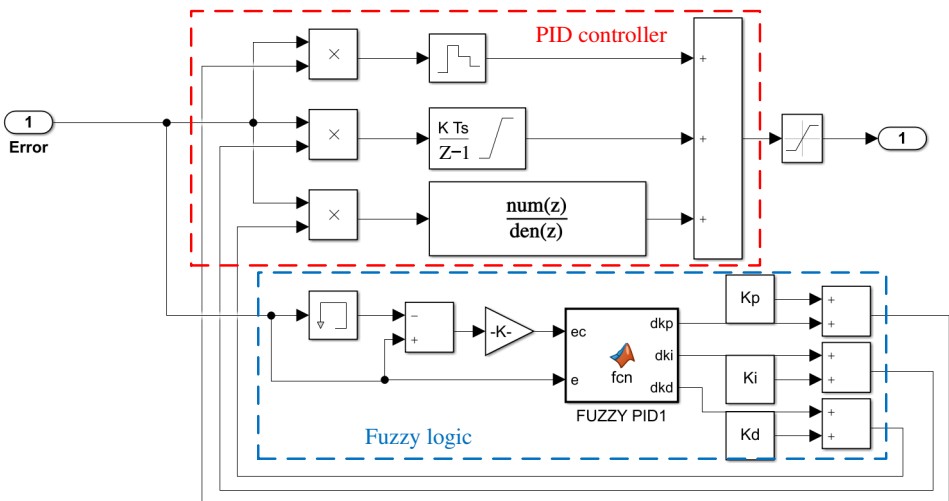

**Figure 7.** Simulation structure of FPID controller.

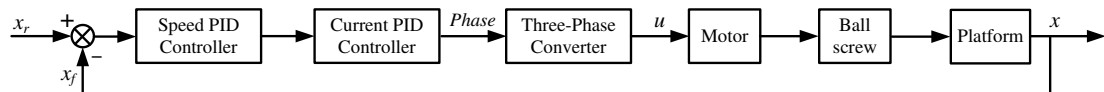

**Figure 8.** PID controller model for Simulink.

In order to verify the tracking ability and anti-interference ability of the controller, a steep response experiment ability of the controller, and a step response experiment, was set in simulation, and the load torque of the motor was increased at 0.6 s to imitate the external interference. As shown in Figure 9, the experimental results show that due to integral saturation, the step response overshoot of the traditional PID controller is the largest, and there is a phenomenon of secondary oscillation. Furthermore, the two kinds of fuzzy PID controller have a smaller overshoot and they eliminate the phenomenon of secondary oscillation, which solves the problem of integral saturation to some extent. For the sudden disturbance, the recovery times of two kinds of FPID controllers are faster, and the maximum disturbance from the set value is smaller, which has a stronger ability to resist external disturbance.

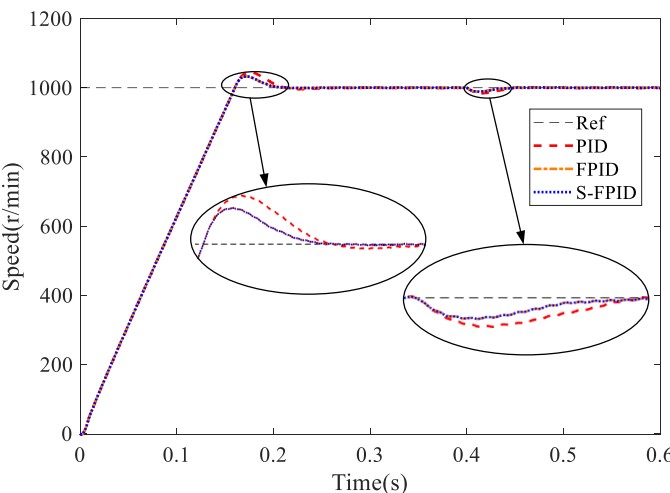

**Figure 9.** Simulation results of step response with disturbance.

The tuning process of parameters of the above three controllers is shown in Figure 10. First, the parameters of the PID controller are fixed, while the parameters of the FPID and S-FPID controllers are changing. Real-time changes in the parameters of the PID controller provide more adjustment capability, which ensures a better controlling performance. Then, when the system is about to reach a stable state and a sudden disturbance is added, the change of parameters of FPID is more complex than those of S-FPID. The complex adjustment of PID parameters is mainly caused by the fuzzy rule base of FPID. Compared with the S-FPID, the fuzzy rule base of FPID has more parameters, which stores more information. More information means more accurate adjustment, which is suitable for controlling the accuracy. However, as observed in Figure 10, the magnitude of the change is very small relative to the current value. The effect of this kind of adjustment on the controlling is limited, which can be neglected for simplification. The simplification of PID parameter determination will improve the efficiency of the FPID controller. Finally, the parameter values of S-FPID and FPID controller are different at the beginning, but over time, they come closer. The similar changing process of the parameter values of the S-FPID controller and the FPID controller, together with almost the same final values, ensures the consistency of the performance of the S-FPID controller and the FPID controller.

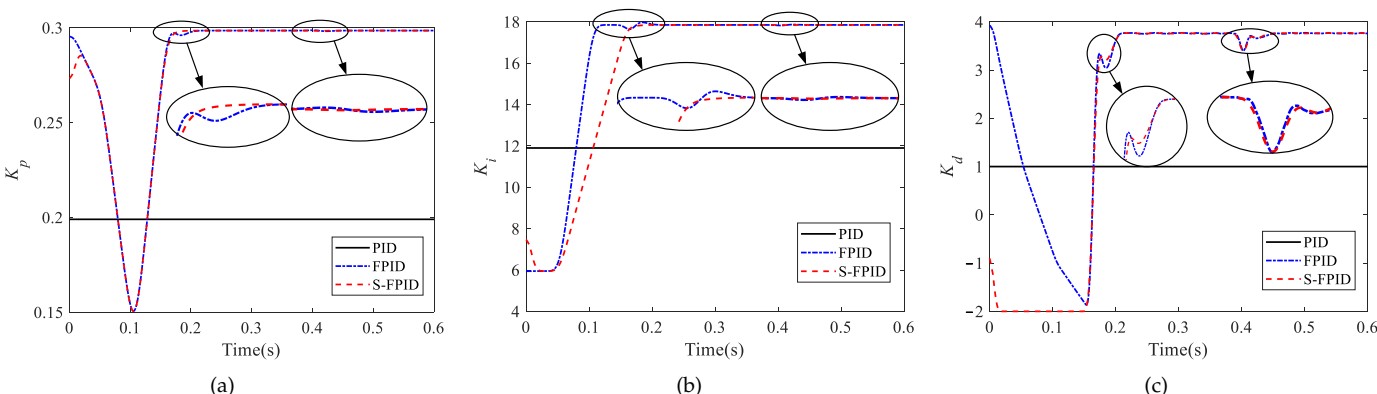

**Figure 10.** The tuning process of parameters: (**a**) $K_p$, (**b**) $K_i$, (**c**) $K_d$.

Based on the discussions above, the proposed S-FPID will have a similar controlling performance to traditional FPID, and have a higher efficiency by neglecting the unnecessary information in the fuzzy rule base. For verification, the performances of the proposed S-FPID controller and the FPID controller are compared through a step response experiment.

It can also be seen from Figure 9 that the performances of the two types of fuzzy controllers are very close. Therefore, the performance of the S-FPID controller has been verified. Compared with the FPID controller, the control effect is the same, but the efficiency has been greatly improved.

Figure 11 shows the output value of the S-FPID (O-S-FPID) controller and the PID controller (O-PID). In the step response, both S-FPID and PID are output with the maximum amplitude limit. After reaching the steady state, the output value of FPID decreases faster and reaches the stable value faster. S-FPID also gives a faster response when the disturbance is suddenly added.

In addition, when the defuzzification algorithms of two kinds of fuzzy PID controllers are run for $1 \times 10^7$ times in MATLAB respectively, the S-FPID controller only needs 0.03 s. Compared with 0.16 s of FPID controller, the running time is reduced by 81.25%, which releases the computational pressure of the CPU greatly. Moreover, the non-zero elements of the sparse fuzzy rule base can be stored using a one-dimensional array alone, and a large amount of RAM space is also saved.

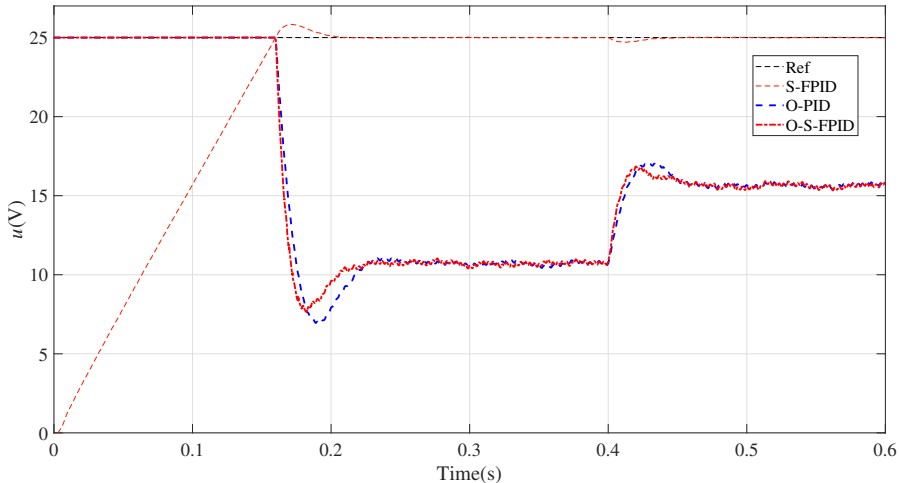

**Figure 11.** Comparison of controller output values under step response.

Compared with the FPID in references [16–18], the S-FPID proposed in this paper has a better operation efficiency due to its structure. As shown in Figure 12, S-FPID deletes the membership degree with weak or even no connection between two inputs, which reduces the fuzzy IF-THEN rules from $n^2$ to $2n - 1$, where $n$ is the number of membership degrees.

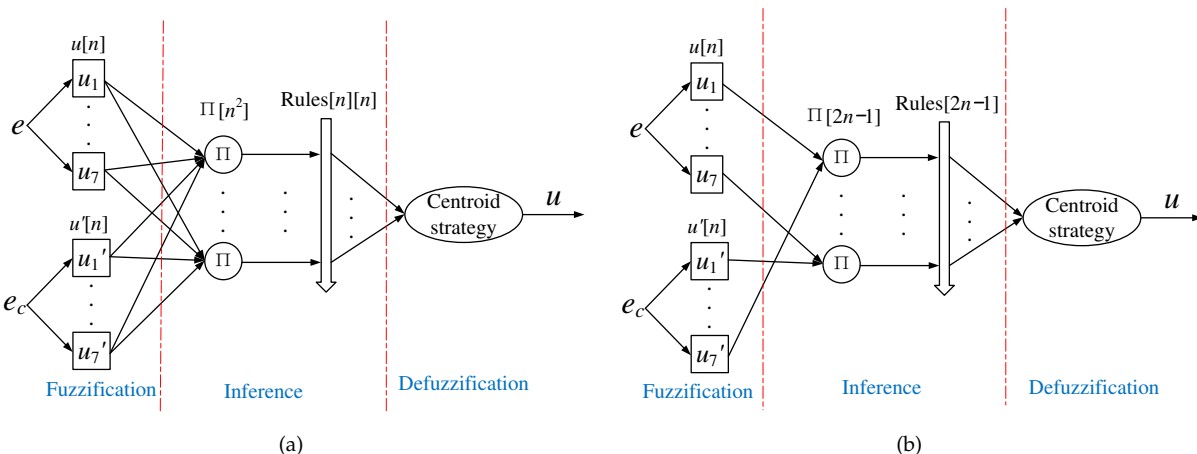

**Figure 12.** Structures of FPID and S-FPID: (**a**) FPID structure, (**b**) S-FPID structure.

In addition to the FPID controller, there are also other intelligent control algorithms such as neural fuzzy control (NFLC) and neural PID controller (N-PID). Figure 13 presents their structures, referring to [20,21], respectively. Compared with S-FPID, the structures of these two controllers are more complex, but they have the ability of parameter learning. The comparison between them is shown in Table 5. When the IF-THEN rule is easy to find and do not require parametric learning ability, S-FPID has advantages in terms of efficiency of operation and robustness. In addition, a simple structure means fewer parameters, which makes it easier for the operators to adjust. An adaptive neuro-fuzzy inference system (ANFIS) composed of such a sparse fuzzy structure and neural network will be equally efficient.

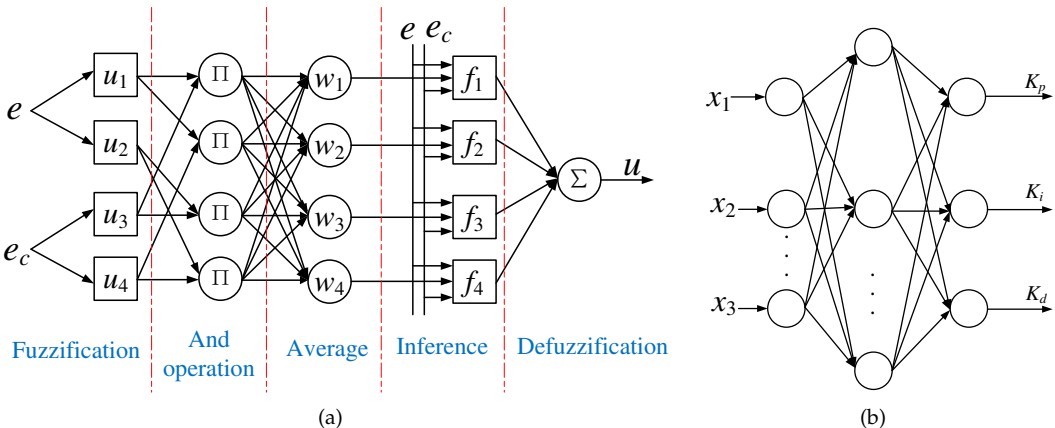

(a)                                                                     (b)

**Figure 13.** Structures of NFLC and neuro-PID: (**a**) NFLC structure, (**b**) N-PID structure.

**Table 5.** Comparison of four kinds of intelligent controller.

| Ability | FPID | S-FPID | NFLC | N-PID |
|---|---|---|---|---|
| Inference | ✓ | ✓ | ✓ | ✗ |
| Learning | ✗ | ✗ | ✓ | ✓ |
| Structure complexity | Complex | Simple | More complex | More complex |

*4.2. Experiment and Analysis*

As shown in Figure 14, the physical experiment devices are mainly composed of a servo motor and its driver, made by the Yaskawa company, ball screw, and platform, etc. In addition, the operation unit of the numerical control system is composed of a PC and a motion control card, in which the motion control card is DMC4080 of GALIL company.

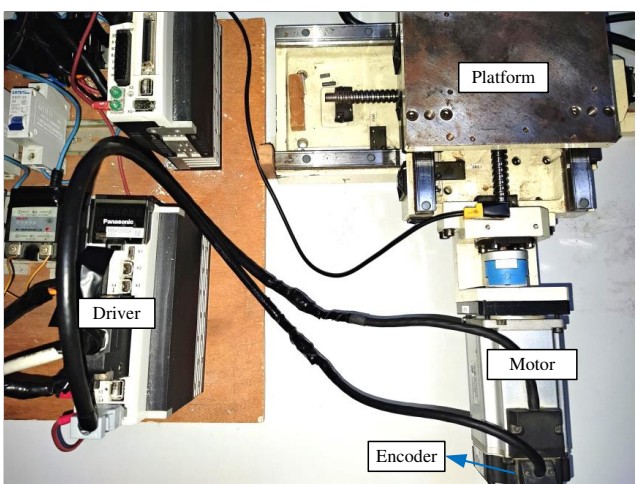

**Figure 14.** Experiment platform.

Figure 15 shows the experimental structure. The PID controller is integrated into the motion control card, and the parameters can be set using instruction code. The fuzzy logic code is written in Labview software using a PC. It can collect the error information of the machine tool in real-time from the motion control card and output the instruction code to change the parameters of the PID controller. Therefore, the FPID controller is composed of a PC and a motion control card, to control the position of the ball screw. The main steps of the experiment are as follows:

1.  Adjust the initial PID controller parameters.
2.  Calculate the PID controller parameter gains using fuzzy rules—refer to Section 3.2.
3.  Calculate $K_p$, $K_i$, $K_d$, and the output value of the PID controller according to Equations (7)–(9).
4.  Collect feedback and determine the position error. Then, repeat steps 2 and 3 to calculate $K_p$, $K_i$, and $K_d$ for the next cycle.

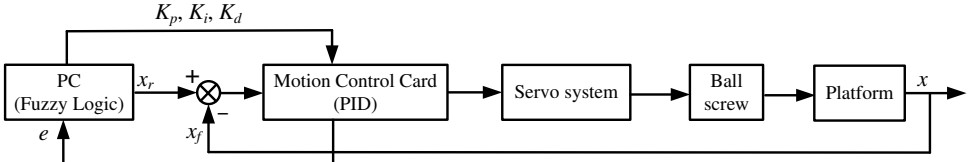

**Figure 15.** Experimental structure.

In order to verify the tracking ability and anti-interference ability of the S-FPID controller, the step response experiment was set to compare with the classical PID and FPID, and the disturbance was manually added to the servo system in the steady state. Before the experiment, the parameters of the PID controller need to be adjusted. The controller parameter values are adjusted to the state with better performance, using the trial and error method. To fairly compare the performance of the three controllers, the initial values of their parameters are the same ($K_p$ = 6, $K_i$ = 0.1, $K_d$ = 1).

As shown in Figure 16, the displacement is represented by the number of coding pulses fed back by the motor encoder (cts). It can be seen that the classical PID controller has the largest overshoot and the weakest anti-interference ability; the FPID controller has the smaller overshoot and the strongest anti-interference ability. The steady-state error and the anti-interference ability of the S-FPID controller is almost unchanged compared with FPID. Although the overshoot is slightly increased, the operation efficiency is significantly improved. Table 6 shows the experimental data of the three types of controllers; compared with the classical PID controller, the overshoot of the proposed S-FPID controller is reduced by 74.29%, and the maximum offset of the sudden increase in disturbance is reduced by 62.32%. In addition, compared with the FPID controller, the S-FPID controller loses 5.71% of the overshoot and 1.67% of the anti-interference performance, but the operation efficiency is improved by 81.25%. The S-FPID controller obtains significant CPU algorithm operation efficiency at a small cost of robustness performance.

**Table 6.** Results.

| Parameters | PID | FPID | S-FPID |
| --- | --- | --- | --- |
| Steady-state error (cts) | 0 | 0 | 0 |
| Overshoot (cts) | 70 | 14 | 18 |
| Recovery time (0.01 s) | 0.3 | 0.3 | 0.26 |
| Offset under disturbance (cts) | 69 | 25 | 26 |
| Fuzzy logic runtime (s) | / | 0.16 | 0.03 |

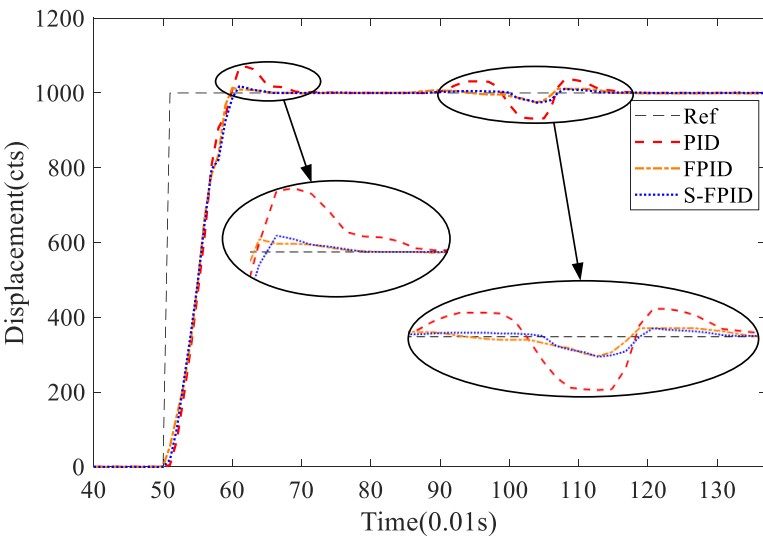

**Figure 16.** Experimental results of step response with disturbance.

## 5. Conclusions

In this paper, the S-FPID controller is proposed to solve the position-tracking problem of a nonlinear servo system, which has the problem of internal parameters and external disturbance uncertainties. The main conclusions are as follows:

1.  FPID solves the integral saturation problems to a certain extent, and it has a stronger ability to resist external disturbance.
2.  The sensitivity analysis method based on MATLAB is an effective means for verifying the adaptive ability of S-FPID controller parameters, which lays a foundation for the success of the real experiment.
3.  Replacing the full-order fuzzy rule base with a sparse rule base can significantly improve the computational efficiency of the CPU and save a lot of RAM space.

The simulation and experimental results show that S-FPID has better tracking performance and anti-interference ability than PID, and that the performance of the S-FPID controller is basically unchanged but has a better efficiency than the FPID controller.

The proposed S-FPID controller can also be applied to other nonlinear motion control systems with a high sampling rate, and process control with a low sampling rate, where the integral saturation phenomenon is easy to occur.

**Author Contributions:** Conceptualization, X.S. (Xianjun Sheng) and Z.Y.; methodology, Z.Y.; software, Z.Y.; validation, K.W., N.L. and X.S. (Xianjun Sheng); formal analysis, Z.Y.; investigation, Z.Y.; resources, X.S. (Xianjun Sheng) and N.L.; data curation, Z.Y.; writing—original draft preparation, N.L. and X.S. (Xianjun Sheng); writing—review and editing, N.L. and X.S. (Xianjun Sheng); visualization, X.S. (Xianghan Sun); supervision, X.S. (Xianjun Sheng); project administration, X.S. (Xianjun Sheng). All authors have read and agreed to the published version of the manuscript.

**Funding:** This research received no external funding.

**Institutional Review Board Statement:** Not applicable.

**Informed Consent Statement:** Not applicable.

**Data Availability Statement:** The data presented in this study are available upon request from the first author.

**Acknowledgments:** The authors thank the DaLian University of Technology (DLUT) for their support.

**Conflicts of Interest:** The authors declare no conflict of interest.

## Abbreviations

The following abbreviations are used in this manuscript:

| | |
|---|---|
| S-FPID | Fuzzy PID controller based on sparse fuzzy rule base |
| CNC | Computer numerical control |
| PID | Proportion integral differential |
| FPID | Fuzzy PID |
| NDOBC | Nonlinear disturbance observer-based control algorithm |
| FSMC | Fuzzy sliding mode control |
| NFLC | Neuro-fuzzy logic control |
| PSU | Pumped storage unit |

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
