# Peer review of "Design of Fuzzy PID Controller Based on Sparse Fuzzy Rule Base for CNC Machine Tools"

_machines, doi:10.3390/machines11010081_

Round 1

Reviewer 1 Report

The paper discusses the development of a FLC in a PID form based on sparce fuzzy rule base. The introductory part of the paper gives some insight into a CNC machinery, and expectations towards the control system. The references in this part are cited in one shot which does not allow the authors to characterize them one by one, see [5-7] or [8-10]. This should be improved. The authors discuss multiple aproaches to deployment of FLCs eventually focusing on a sparse FLC one. No clear novelty, neither contribution statements are given. 

Next, the authors present a servo model with a moving platform to exert linear motion. Subsequently, the uncertainty of the model is taken into account, supporting the use of a FLC. Below (8) it is not stated whether the updates in xe or xec are with the same frequency as the updates in (9). Next, the PID controller model is unrealistic, as it does not take control limits into account, windup avoidance strategy or moving D action to the feedback term. This should be better supported. Please refer to Figure 7 when this is actually taken into account

What is the impact of gain tuning on the control algorithm itself? Usually when the error is large, it is the control action itself to be taken solely, not gain changes. 

IN the experimental section - what is the tuning method for parameters of PID and FPID? Please support it giving precise criteria and showing the control signals actually behave alike, to make the results comparable. Otherwise, it is easy to show the superiority of your approach. Any standard robust control approaches to present? 

I believe the structure of the paper, support for this particular approach, alternative approaches, experimental report section, should be deeply updated. 

Reviewer 2 Report

The author presents the article entitled “Design of Fuzzy PID Controller Based on Sparse Fuzzy Rule Base for CNC Machine Tools”

 This paper proposes a fuzzy PID controller based on a sparse fuzzy rule base (S-FPID), which can effectively control the position of a nonlinear CNC machine tool servo system consisting of a rotating motor and ball screw. The article is well-structured and easy to read. However, It presents the following concerns:

The article presents the following concerns:

  • It is recommended to give a brief background of the work in the Abstract section.

  • Figures must be vectorized.

  • Line 37, regarding nonlinear problems with PID controllers, can be justified with the reference: A PID-type fuzzy logic controller-based approach for motion control applications; Auto-regression model-based off-line pid controller tuning: an adaptive strategy for dc motor control.

  • I suggest adding a Discussion section to discuss the results and how they can be interpreted from the perspective of previous studies and of the working hypotheses. It is also recommended to add a table that compares the main findings of the work vs the already reported in the literature.

  • I recommend a comparison table by considering the findings reached by the state-of-the-art.

Reviewer 3 Report

In the article under review, the authors presented the results of a study of a fuzzy PID controller based on a sparse fuzzy rule base (S-FPID) applied to a servo drive consisting of a motor and a ball screw. The studies were carried out by simulation using the Matlab/Simulink software, as well as on an experimental laboratory platform. The authors compared the efficiency of functioning of the known FPID and PID controllers and the S-FPID controller recommended by them.

In the Introduction, the prerequisites for conducting research are considered in sufficient detail, and the purpose and objectives of the paper are formulated. In the main parts of the paper, a description of the dynamic mathematical model of the servo system is presented. A description of the FPID controller and a detailed description of the S-FPID controller are given. The results of the simulation and the results of a physical experiment on a laboratory setup are shown. Conclusions are drawn with recommendations for the use of the S-FPID controller for different non-linear motion control systems.

In my opinion, the results of the study have good practical importance. At the same time, in their research, the authors only applied previously well-studied control algorithms to a well-known technical object, which is not of high scientific value. I think, in the presented form, the paper should be presented at the conference and can be published as conference proceedings. However, for publication in a highly-rated journal, the paper needs major revision, namely, it is necessary to clearly formulate and describe in detail the scientific novelty of the research presented in the paper.

Round 2

Reviewer 1 Report

The authors have updated the paper according to my comments, however I cannot see any deep revision. 

I cannot agree that sparse FLC is well supported, and related to the literature, including deep discussion between this and alternative approaches. This should take into account the 3 approaches from Fig. 9, and their tuning process which would enable one that we compare equal with equal approaches. 

The experiments are also not well described, not allowing the other researchers to reproduce the results. 

Reviewer 3 Report

In the new version of the paper, my recommendations were taken into account. The article in this version can be accepted.

Author Response

Thank you again for your valuable suggestions, which are of great help to improve the quality of our paper.

Round 3

Reviewer 1 Report

Thank you for taking my comments into consideration. I believe the paper is publishable in its current form now. Good luck with the review process!